# *Ianus Bifrons*: The Two Faces of Metformin

**DOI:** 10.3390/cancers16071287

**Published:** 2024-03-26

**Authors:** Umberto Goglia, Iderina Hasballa, Claudia Teti, Mara Boschetti, Diego Ferone, Manuela Albertelli

**Affiliations:** 1Endocrinology and Diabetology Unit, Local Health Authority CN1, 12100 Cuneo, Italy; 2Endocrinology Unit, IRCCS Ospedale Policlinico San Martino, 16132 Genova, Italymara.boschetti@unige.it (M.B.); ferone@unige.it (D.F.); manuela.albertelli@unige.it (M.A.); 3Endocrinology Unit, Department of Internal Medicine and Medical Specialties (DiMI), University of Genova, 16132 Genoa, Italy; 4Endocrinology and Diabetology Unit, Local Health Autorithy Imperia 1, 18100 Imperia, Italy; c.teti@asl1.liguria.it

**Keywords:** biguanide, metformin, glucose control, diabetes, cancer treatment, clinical trials

## Abstract

**Simple Summary:**

Metformin is one of the most widely used drugs in the world, since its first use in the second part of the 1950s in France (Aron, 1957) and in the United Kingdom (Rona, 1958), and is on the World Health Organization’s List of Essential Medicines. Traditionally, metformin is used in diabetes mellitus, mainly in overweight and obese subjects, but also in other conditions of impaired glucose metabolism, such as insulin resistance. In recent years, in vitro and in vivo data on this molecule as an antiproliferative modulator have suggested a series of clinical trials in several tumors that—at the moment—have not confirmed the expected positive results. The aim of this paper is to offer a double vision of this molecule, with a very light side about the metabolic aspects and a darker side concerning its antineoplastic activity, focusing on molecular and subcellular pathways involved in both fields and discussing the contrast between preliminary cellular data and real clinical outcomes in different neoplasms.

**Abstract:**

The ancient Roman god *Ianus* was a mysterious divinity with two opposite faces, one looking at the past and the other looking to the future. Likewise, metformin is an “old” drug, with one side looking at the metabolic role and the other looking at the anti-proliferative mechanism; therefore, it represents a typical and ideal bridge between diabetes and cancer. Metformin (1,1-dimethylbiguanidine hydrochloride) is a drug that has long been in use for the treatment of type 2 diabetes mellitus, but recently evidence is growing about its potential use in other metabolic conditions and in proliferative-associated diseases. The aim of this paper is to retrace, from a historical perspective, the knowledge of this molecule, shedding light on the subcellular mechanisms of action involved in metabolism as well as cellular and tissue growth. The intra-tumoral pharmacodynamic effects of metformin and its possible role in the management of different neoplasms are evaluated and debated. The etymology of the name *Ianus* is probably from the Latin term *ianua*, which means *door*. How many new doors will this old drug be able to open?

## 1. Introduction


*…cur de caelestibus unus*



*Sitque quod a tergo sitque quod ante vides.*





*Tum sacer ancipiti mirandus imagine Ianus*


*bina repens oculis obtulit ora meis* [1].

In ancient Italian and Latin religions, *Ianus* is considered a singular divinity with mysterious features. In fact, he is the god of material and immaterial primordial times in human history, and his figure is strictly connected to the concept of time.

The picture of Ianus shows a double facial image in opposite vision, one to the *past*, the other to the *future* (Figure 1).

In the divinity pantheon, Ianus occupies a very relevant and superior seat; indeed, he is also named *Divus Deus* (“God of Gods”), *Divum Pater* (“Father of the Gods”), and *Ianus Bifrons* (“God with two faces”), and he is also the only idol without his own parents.

Moreover, in a mythological vision of the ancient history of Rome, Ianus was the first king of the land of Latium, founding the ancient town on top of Gianicolo Hill, receiving the feature of the vision of the past and of the future from the god Saturnus.

Just like the Father of Gods, metformin is an old, ambivalent molecule, presenting at least a double pathophysiological connection, looking to metabolic effects and to antiproliferative actions.

### 1.1. Historic Perspective

In 1772, the famous botanist Sir John Hill described the *Galega Officinalis* as “Perennial, native of Spain and Italy; of Greece and Africa, a specious plant, of a yard high, that flowers in August. The Stalk is juicy, and green… the Flowers are purples; sometimes white” (Figure 2). This plant—also named goat’s rue, French lilac, Italian fitch, Spanish sainfoin, professor weed, or *Herba routae caprariae*—was used in medieval Europe as a traditional cure for worms, epilepsy, fever, and pestilence and to treat conditions of thirst and frequent urination [2,3].

In the nineteenth century, the works of Adolf Strecker and Bernhard Rathke, with the preparation of guanidine and the fusion of two guanidine molecules to form biguanides, respectively, were determinants [4,5]. Therefore, the synthesis of metformin (dimethyl-biguanide) was carried out in 1922 by Wermer and Bell, showing lower toxicity than the other mono- and diguanidines [6,7]. Later, an independent research line about antimalaric agents documented that metformin caused a lowering of blood glucose in animal studies [8,9]. The most persevering and passionate scientist who dedicated his own life to the study of metformin was Jean Sterne, a physician at the hospital in Casablanca and later at Aron Laboratories in Suresnes, in the west of Paris, France [10]. Between 1957 and 1958, Stern showed that N,N-dimethylamine guanyl guanidine (metformin) had a blood sugar-lowering effect, replacing the need for insulin in subjects with a relative deficiency of insulin [11,12,13]. To market metformin, Aron adopted the trade name ‘glucophage’ (from the ancient Greek language, meaning *glucose eater*); later, Jean Stern played a prominent role in further research and physician education to assist the introduction of metformin into clinical practice in Europe [14,15,16,17,18]. Metformin, unlike the other biguanides (mainly phenphormin, removed from the market in the USA in 1978), showed a more favorable safety profile, with distinct and specific differences in pharmacokinetic features [19,20].

### 1.2. Chemical and Pharmacological Features

The chemical structures of guanine, guanidine, and biguanides are shown in Table 1. Metformin (1,1-dimethylbiguanide hydrochloride) is a relatively planar hydrophilic molecule, monoprotonated at neutral pH with several tautomeric configurations.

**Absorption**. After the oral intake (approximately 35 mg/kg/day, that is 500–2550 mg/day), metformin is absorbed mainly in the small intestine cells (20% duodenum, 60% jejunum, and ileum, 20% colon). After 1.5 h is the onset of action, about 1.5–4.9 h is the half-life in circulation, and 16–20 h is the duration of action. Metformin hydrophilicity represents a limiting step of oral absorption due to the low intestinal permeability [21,22,23].

**Distribution**. The distribution of metformin is rapid and without binding to plasma proteins [24,25]. In hepatic tissue, the concentration is three- to five-fold higher than in the portal vein (40 µmol/L), so the hepatocyte is the primary site of drug function [26]. Generally, within 24–48 h, steady-state plasma concentrations are reached (approximately <1 mcg/mL).

The presence and expression of transporters for cationic compounds (OCT), plasma membrane monoamine transporter (PMAT), and multidrug and toxin extrusion proteins (MATE) are critical for the biodistribution and pharmacodynamics of the biguanide [27,28,29,30,31] (Figure 3).

**Metabolism and elimination**. Metformin, not metabolized, is present unchanged in the urine, with a half-life of about 5 h. In the kidney, active tubular secretion is the main route of removal, with a plasma elimination rate of about 500 mL/min [22,32]. Low molecular weight, absence of plasma protein binding, presence of renal transporters, and low lipid solubility are the main factors inhibiting passive reabsorption. The impairment of renal function reduces the clearance of drugs [33].

**Therapeutic range**. The therapeutic range of metformin is unclear, but it should not exceed 5 mg/L [22,34]. Both formulations of metformin (immediate and extended release) displayed similar areas under curve (AUC) and equal safety profiles and efficacy [22]. Lactic acidosis is the most dangerous potential risk, so in critical cases and frail subjects, an accurate monitoring should be performed, maintaining a drug concentration below 2.5 mg/L [25].

## 2. Metformin Target Organs

### 2.1. Liver

The liver and skeletal muscle have been considered the major target organs of metformin action for many years, while, more recently, other sites of action have been highlighted for an important role beyond simple glycemic control, among these: the gastrointestinal tract, intestinal microbial communities, and tissue-resident immune cells. The pharmacodynamics, even at the molecular level, appear to be influenced by the dose and duration of treatment with metformin [35]. Metformin has been defined as an insulin sensitizer, leading to a reduction in insulin resistance and therefore optimizing cellular glucose uptake, mainly in skeletal muscle, leading to a reduction in plasma glucose and insulin plasma values. This effect of metformin could be attributed to its positive effects on insulin receptor expression and tyrosine kinase activity on the phosphorylation of insulin receptors [36]. Most studies in the literature conclude that the main function through which metformin reduces hepatic glucose production is the inhibition of gluconeogenesis [37,38]. Inhibition of gluconeogenesis has been explained by both demonstrating changes in enzymatic activity such as pyruvate kinase flux and modifications of citric acid cycle activity and also highlighting a reduction in hepatic uptake of gluconeogenic substrates [39,40,41,42].

### 2.2. Gastrointestinal System

More recently, the gastrointestinal tract has been identified as an additional site of action for metformin; in fact, a high accumulation of metformin in the intestine has been reported, both in humans and in animal models, with concentrations up to a couple of hundred times higher than those of plasma and other tissues, suggesting that the intestine acts as an important reservoir of metformin. Metformin increases glucose absorption in the basolateral intestine and, through gut–liver communication, influences hepatic glucose production [43,44,45,46]. Several findings suggest that the hypoglycemic capacity of metformin depends both on intestinal glucose absorption along the gastrointestinal tract and on bloodstream absorption. After entering enterocytes, anaerobic glucose metabolism causes the accumulation of lactate and acetate in the wall of the small intestine and its release into the circulation, creating an intestinal–liver communication to attenuate gluconeogenesis [44,47,48,49]. Metformin can therefore exert its beneficial metabolic actions through modulation of the incretin axis, enhancing circulating levels of glucagon-like peptide 1 (GLP-1), inducing glucagon-like peptide-1 (*GLP-1*) *receptor* gene expression, and reducing plasma dipeptidyl peptidase-4 activity through a mechanism that depends on PPAR (peroxisome proliferator-activated receptor)-α [50,51].

### 2.3. Brown Adipose Tissue

An increasing number of studies have shown that BAT, in addition to having a role in dissipating energy through heat production, contributes to the regulation of glucose homeostasis [52,53]. The presence of metformin has been demonstrated in the interscapular BAT of mice using 11C-metformin PET imaging, which supports the hypothesis that BAT could be another important target of metformin [54]. At the BAT level, metformin also appears to be involved in lipid metabolism; in fact, by increasing the activity of hormone-sensitive lipase and AMP-activated protein kinase (AMPK), metformin promotes intracellular triglyceride uptake, lipolysis, and subsequent mitochondrial fatty acid oxidation [55,56].

## 3. Molecular Mechanisms Involved in Metformin’s Action

### 3.1. Metformin-Induced Reduction in Blood Levels of Glucose (“Metabolic Face”)

The main mechanisms producing the anti-hyperglycemic action of metformin reside in mitochondria and lysosomes, after the interaction of metformin with organic cationic transporter 1 [57,58].

Organic Cation Transporter 1 (OCT1), a member of the family of membrane transporters named Solute Carrier 22 (SLC22), facilitates the movement of endogenous and exogenous compounds across cell membranes and is primarily localized on the basolateral membrane of hepatocytes. Metformin is a well-known substrate of OCT1, and genetic polymorphisms of OCT1 are known to reduce the effectiveness of metformin by lowering absorption and causing gastrointestinal intolerance to metformin [59,60,61]. A recent paper by Zeng et al., using cryo-electron microscopy, showed the interaction between drug and OCT1 in different conditions, substrate-free and substrate-bound, with a resolution of 3.5 Å. Conformational changes from outward-to inward-facing states are demonstrated for the first time in a very suggestive way [58].

#### 3.1.1. Complex I Inhibition-Dependent Mechanism

Mitochondrial Complex-I (NADH: ubiquinone oxidoreductase) is a crucial component for respiration in aerobic organisms, oxidizing NADH from the tricarboxylic acid cycle and β-oxidation. In hepatocytes, metformin induces a reversible inhibition of mitochondrial respiratory chain complex I, as documented by studies combining cryo-electron microscopy and enzyme kinetics [62,63]. After the intracellular uptake of metformin, three main phenomena demonstrated in hepatic cells are a higher NADH/NAD^+^ ratio, a reduction in ATP concentration, and increased levels of AMP.

#### 3.1.2. Mitochondrial Glycerol-3-Phosphate Dehydrogenase (mGPDH)-Dependent and Complex IV Inhibition-Dependent Mechanism

Complex-IV is a cytochrome c oxidase involved in the final steps of energy conservation. At the hepatic level, metformin acts directly and indirectly via Complex-IV, inhibiting mGPDH, with the following results: a higher NADH/NAD^+^ ratio, a reduction in gluconeogenesis from lactate, and a reduction in the activity of the glycerol–phosphate shuttle, which transfers NADH from the cytosol to mitochondria [64,65]. Finally, the hepatic redox state is raised through an increase in the glutathione to oxidized glutathione ratio (GSH:GSSG), with inhibitions of genes involved in the process of gluconeogenesis.

#### 3.1.3. AMPK Activation-Dependent Mechanisms in Lysosomes

AMP-activated protein kinase (AMPK) is a master controller of metabolic homeostasis. At low concentrations, metformin binds presenilin enhancer 2 (PEN2), which is recruited to ATPase H+ transporting accessory protein 1 (ATP6AP1) independent of changes in AMP levels, leading to inhibition of v-ATPase and phosphorylation/activation of AMPK through the formation of a supercomplex containing the v-ATPase, AXIN, liver kinase B1 (LKB1), and AMPK [66]. Metformin-activated AMPK from lysosomes reduces lipid accumulation in the liver via acetyl-CoA carboxylase (ACC) inhibition and increases glucagon-like peptide 1 (GLP1) secretion in the gut, inducing reductions in blood levels of glucose.

### 3.2. Anticancer Molecular Mechanisms of Metformin (“Anti-Proliferative Face”)

Accumulating data from preclinical studies support the anti-neoplastic activity of metformin in several malignancies, thus providing the rationale for further exploration of the biguanide in more than 130 clinical trials [67]. Nevertheless, its underlying molecular mechanisms have not been fully clarified. To date, metformin anti-carcinogenic impact has been generally classified as direct, i.e., glucose- and insulin-independent, or indirect, i.e., glucose- and insulin-dependent, both activities not mutually exclusive [68] (Figure 4).

#### 3.2.1. Direct Anticancer Mechanisms of Metformin

##### The mTOR Pathway

Cancer cells, in comparison to normal ones, present an aberrant metabolism with higher requests for catabolite uptake and utilization, which are necessary for survival and growth. Reprogramming cellular energy homeostasis, in fact, represents one of the essential mechanisms through which metformin appears to directly attenuate tumorigenesis and progression [68]. A crucial anti-tumor target of the biguanide concerns the mechanistic target of rapamycin complex 1 (mTORC1), which is a major driver of protein biosynthesis, cell growth, and metabolism as a response to different stimuli such as growth factors, nutrient availability, energy, and oxygen intracellular levels [69]. In particular, metformin selectively inhibits mitochondrial respiratory-chain complex 1 (NADH coenzyme Q oxidoreductase), ultimately leading to decreased cell respiration, reduced oxidative phosphorylation (OXPHOS), and ATP depletion. Metformin-induced energy stress triggers the activation of AMP-kinase (AMPK), which plays a key role in modulating critical pathways such as mTORC1 signaling. Specifically, AMPK suppresses mTORC1 activity directly via phosphorylation of S722 and S792 on Raptor, its scaffolding protein, as well as through the activation of the tuberous sclerosis complex (TSC). The inhibitory impact of TSC1 and TSC2 is also exerted on the mTORC1 downstream major effectors such as eukaryotic initiation factor 4E-binding protein 1 (4EBP1) and ribosomal protein kinase S6 (S6K) [68]. Furthermore, metformin may suppress mTORC1 activity in an AMPK-independent way by inhibiting Rag GTPases, essential for the amino acid signaling to mTORC1, as well as by promoting the activation during hypoxic stress of REDD1 (regulated in development and DNA damage responses), a hypoxia-inducible factor-1 target gene involved in cell survival regulation [70,71,72].

The anticancer activities of the biguanide via repression of the mTORC1 pathway, as above-mentioned, were documented in several preclinical studies performed in different malignancies such as lung cancer, pancreatic cancer, prostate cancer, breast cancer, thyroid cancer, meningioma, leukemia, and lymphoma [73,74,75,76,77,78,79,80].

##### The PI3K/AKT/mTOR Pathway

Another anticancer metformin mechanism of action involves the PI3K/AKT/mTOR (PAM) pathway, which is a major signaling network modulating cell growth, metabolism, proliferation, as well as apoptosis and autophagy [68,81]. The overactivation of the PAM axis represents one of the main drivers of tumor pathogenesis and progression, as well as of anti-tumor therapeutic resistance. Several preclinical studies in different cancer types have suggested that metformin may exhibit its antiproliferative effects through the inhibition of PI3K/AKT/mTOR. In bladder cancer cells, the biguanide in a concentration-dependent manner led to the reduction of PI3K, AKT, and mTOR phosphorylation and was ultimately associated with the suppression of cell proliferation and migration, the activation of the caspase cascade, and the induction of apoptosis [82]. In a study performed by Nozhat et al. in anaplastic thyroid cancer (ATC) cell lines, metformin in a time- and dose-dependent way repressed cell growth, significantly altered ATC cellular morphology, and decreased cell migration, likely by reducing the mRNA expression of PI3K and AKT, with no impact, however, on their phosphorylation status [83]. Tang and colleagues demonstrated the antiproliferative activity of biguanide also in esophageal cancer cells, mediated partly by the suppressed expression of the insulin-like growth factor 1 receptor (IGF-1R) and its downstream targets PI3K, AKT, mTOR, p70S65, and PKM2 [84]. Furthermore, metformin not only inhibited cervical cancer cell survival and proliferation, but also increased NK cell cytotoxicity by modulating through the PI3K/AKT axis the expression of MICA and HSP70 proteins on the cellular surface [85]. In colorectal cancer cells (CRCs), biguanide exhibited its inhibitory effect on CRC growth by down-regulating the PI3K/AKT pathway and repressing the expression of inhibin BetaA (INHBA), an oncogene member of the TGF-beta family [86]. Finally, the antiproliferative activity of metformin mediated by the interference with the PAM axis was also documented in preclinical studies conducted in ovarian, endometrial, and breast cancer, hepatocellular carcinoma, and mouse melanoma B16 cells [87,88,89,90].

##### The K-Ras Pathway

A novel anti-tumor target of metformin is the K-Ras pathway, which plays a crucial role in cell signal transduction, differentiation, and proliferation. Aberrant K-Ras contributes to oncogenesis and is frequently associated with poor outcomes and resistance to anti-EGFR therapy. Several studies in vitro and in vivo documented the inhibitory effects of the biguanide on the growth of K-Ras-mutated cancer cells. In particular, in endometrial cancer models, metformin inhibited cell proliferation and triggered apoptosis in a concentration-dependent manner by displacing the oncogenic K-Ras from the plasma membrane with the subsequent suppression of its biological activity, as well as by down-regulating downstream MAPK signaling [91]. In K-Ras-mutated CRC, metformin exerted antiproliferative effects by inactivating both the RAS/ERK and AKT/mTOR pathways [92]. Moreover, in K-Ras mutant lung adenocarcinoma and pancreatic cancer cell lines, biguanide, in a dose-dependent manner, induced apoptosis and suppressed cell proliferation by targeting crucial downstream effectors of K-Ras signaling such as MAPK and AKT [93].

##### The NKL Pathway

The anti-tumor activity of metformin is also exerted by interfering with nemo-like kinase (NLK), a member of the MAPK family, which carries out a relevant role in cell cycle progression and contributes to the oncogenesis of several neoplasms, including colon, prostate, lung, and hepatocellular cancer. Data in vitro and in vivo in non-small-cell lung cancer (NSCLC) cell lines showed that the biguanide—via inhibition of NLK expression—induced cell cycle arrest and significant reduction of the stem cell tumor population [94].

##### The JNK Pathway

Metformin may exert anti-proliferative activity also by interfering with c-Jun-N-terminal kinase (JNK, also known as stress-activated protein kinase 1—SAPK1) signaling, another MAPK-involved pathway that regulates cell growth, survival, proliferation, and migration [95]. In osteosarcoma cell lines, the biguanide activates the JNK cascade and promotes cell cycle arrest and programmed cell death processes, including apoptosis and autophagy [96]. Similarly, in lung cancer cell lines, metformin led to increased apoptosis and suppression of cell proliferation in a dose- and time-dependent manner, either by activating JNK/p38 MAPK signaling or upregulating the expression of DNA damage inducible gene 153 (GADD153) [97]. In gastric adenocarcinoma cell lines, biguanide was associated with remarkable anti-proliferative activity and apoptosis induction also through the phosphorylation reduction of several MAPKs such as JNK, ERK, and p38, in addition to the activation of AMPK and repression of the AKT/mTOR pathway [98].

Moreover, He et colleagues documented the inhibitory effects of metformin on the viability of thyroid cancer TPC-1 cells via down-regulation of LRP2, a transmembrane receptor mainly involved in lipid metabolism, ultimately leading to the suppression of JNK signaling in a concentration-dependent way [99].

##### The STAT3 Pathway

Another pathway involved in the anti-neoplastic effects of metformin concerns STAT3 (signal transducer and activator of transcription 3), which is a promising cancer therapeutic target due to its crucial role in cell survival, proliferation, and migration. Preclinical studies suggest that the biguanide activity is mediated by reducing STAT3 phosphorylation and/or suppressing its nuclear translocation, thus also leading to the downregulation of its target genes such as cyclin D1, Bcl-XL, and Bcl2. Cyclin D1 is a vital regulator of cell cycle progression, which promotes cell cycle G1/S phase transition, while Bcl-XL and Bcl2 are pro-survival genes directly transcribed by STAT3 and involved in programmed cell death modulation [100]. In fact, by repressing the STAT3 pathway in in vitro and in vivo models, metformin inhibited cell proliferation and induced apoptosis and autophagy in a dose- and time-dependent manner in several cancer types, including esophageal squamous cell carcinoma, triple-negative breast cancer, endometrial, ovarian, bladder cancer, glioblastoma, chronic neutrophilic leukemia with mutated CSF3R, and cholangiocarcinoma [101,102,103,104,105,106,107]. Furthermore, in bladder cancer models, metformin also suppressed cell migration and invasion, ultimately interfering with the progression of precancerous lesions. The underlying mechanism of action suggested is the biguanide-induced inhibition of STAT3 signaling, which regulates the activity of matrix-metalloproteinases (MMPs), Rho, and Rac proteins that carry out a key role in modulating cellular migration [108]. Additionally, in primary breast cancer cells, high doses of metformin attenuated cancer progression via suppression of the STAT3 and NF-kB pathways. In particular, the biguanide inhibited the IL-6 epithelial–mesenchymal transition (ETM) and decreased the expression of mesenchymal markers, which are crucial for tumor metastasis [109].

##### The HER2 Pathway

Metformin is also associated with anticancer activity in HER-2 positive cancer via interference with human epidermal growth factor receptor-2 (HER-2) signaling. HER2 is a relevant oncogene able to modulate several key genes such as TP53, CDK12, PI3KCA, and PTEN, which furthermore contribute to tumor aggressiveness and progression [110,111,112]. In particular, in the study performed by Vazquez-Martin on breast carcinoma (BC) cells, biguanide in a dose- and time-dependent manner led to a significant reduction of HER-2 expression, mainly by inhibiting the downstream effector of mTOR p70S6K1, also in an AMPK-independent way [113]. In another preclinical model of HER-2 BC cells, metformin inhibited cell proliferation and induced apoptosis, likely due to the inhibition of heat shock protein 90 (HSP90), which downregulates the AKT and MAPK pathways [114].

Moreover, metformin may abrogate HER-2-induced tumor angiogenesis via targeting the HER2/HIF-1α/VEGF pathway and is associated in a dose-dependent manner with inhibition of HER2-positive gastric cancer cell growth, also via reduction of HER 2 phosphorylation [111,115].

##### The NF-κB Pathway

Other antitumor effects of metformin are mediated by interference with the nuclear factor κB pathway (NF-κB), which is involved in cancer development, invasion, and metastasis via modulation of EMT, as well as in therapeutic resistance [116]. In the study of Li and colleagues, conducted in EGFR-mutant lung cancer models with acquired resistance to EGFR tyrosine kinase inhibitors (TKIs), the biguanide inhibited cell proliferation, induced apoptosis, reversed or delayed the TKIs resistance, and repressed cancer cell stemness by inactivating ERK/NF-κB signaling in an AMPK-dependent way [117]. Metformin is able to trigger caspase3/GSDME-mediated pyroptosis of cancer cells via stimulation of the AMPK/SIRT1/NF-κB pathway and mitochondrial dysregulation [118]. Furthermore, in primary breast cancer cell lines, metformin presents an inhibitory effect on tumor invasion and metastasis due to the dual suppression of NF-kB activity and nuclear translocation mediated by MMP-9 downregulation [119], and in prostate cancer models, metformin appeared to attenuate metastasis by repressing NF-Kb signaling, thus leading to the suppression of tumor necrosis factor-α-induced EMT [120]. In a preclinical study performed on ovarian cancer cells, biguanide was shown to suppress cancer progression and reduce chemoresistance via modulation of NF-κB and IL-6 signaling [121].

#### 3.2.2. Indirect Anticancer Mechanisms of Metformin

The indirect antineoplastic effects of metformin are exerted by modulating blood glucose and insulin concentration. In fact, the biguanide down-regulates insulin and insulin-binding proteins, leading to decreased levels of insulin growth factor-1 [68]. The IGF axis plays a crucial role in cell growth and metabolism. Moreover, IGF-1 and IGF-2, in particular, promote both mitogenic as well as antiapoptotic signaling via activation of other key pathways such as PI3K/AKT/mTOR and RAS/RAF/MAPK [122]. Additionally, metformin-induced AMPK activation reduces insulin receptor substrate-1 (IRS-1) phosphorylation, which interferes with the PI3K/AKT/mTOR axis too [68]. Finally, in endometrial cancer cells, metformin presented antiproliferative effects by reducing IGF-1 secretion and IGF-1R expression, ultimately leading to the inhibition of downstream PI3K/AKT signaling [123], whereas considering breast tumors, in triple-negative cancer cell lines, a synergistic effect of metformin combined with an insulin/IGF-1 inhibitor in suppressing cell growth and proliferation was documented [124,125].

## 4. Clinical Trials on Metformin Use

### 4.1. Clinical Studies on Hyperglicemic Conditions (“Metabolic Face”)

#### 4.1.1. Type 2 Diabetes Mellitus

The series of United Kingdom Prospective Diabetes Studies (UKPDS) represent the main source about the normoglycemic role of metformin, involving subjects with type 2 diabetes mellitus in intensive and conventional dietary treatment. In overweight individuals taking metformin, reductions in the risk of myocardial infarction of 39% (*p* = 0.01) and of death from any cause of 36% (*p* = 0.01) were observed [126,127,128,129].

Moreover, Holman and colleagues conducted post-trial monitoring of UKPDS, demonstrating that in the overweight group (342 subjects in metformin), compared with subjects in conventional therapy, differences in glycated hemoglobin levels were lost after the first year, but significant risk reductions persisted for several end points. This phenomenon was called the “legacy effect”, the persistence of clinical benefits despite the early loss of within-trial differences in glycometabolic balance [130].

In a systematic review and meta-analysis about diabetes medications as monotherapy or metformin-based combination treatment, Maruthur et al. demonstrated that major hypoglycemic risk and cardiovascular mortality were lower for metformin versus sulfonylureas and that reductions in glycated hemoglobin values were similar across monotherapies and metformin-based combinations, except for DPP-4 inhibitors, which showed a smaller effect. Metformin, DPP-4 inhibitors, and GLP-1 receptor agonists were similar in reducing or maintaining body weight, which increased with sulfonylureas, thiazolididinediones, and insulin (differences up to 5 kg). As expected, genital infections were increased with glifozins, while gastrointestinal adverse events were highest with metformin and GLP-1 receptor agonists [131].

Furthermore, it is also important considering the pleiotropic actions of metformin and the worsening of neuropathic symptoms in subjects with vitamin B12 deficiency due to the chronic assumption of biguanide, so periodic testing of vitamin B12 is recommended [132,133].

Several institutional guidelines do not suggest metformin as a first-line treatment in secondary cardiovascular prevention, lacking protective impact on major adverse cardiovascular events, cardiovascular death, myocardial infarction, heart failure, and stroke, as demonstrated in an umbrella review of a group from Sun Yat-sen University [134].

#### 4.1.2. Pre-Diabetes

The main projects aimed at analyzing subjects at very high risk of developing diabetes were the Diabetes Prevention Program (DPP) at the end of the last century (1996–2001) and later the DPP Outcome Study (DPPOS, 2002–2013). These trials demonstrated that intensive lifestyle treatment and biguanide therapy were favorable vs. placebo in preventing the onset of diabetes (7.0%, 5.7%, and 5.2% per year for placebo, metformin, and lifestyle, respectively) and 27% and 18% lower for lifestyle and metformin vs. placebo, respectively [135].

In a very recent review, Patel and colleagues—in a total sample size of 30,474 subjects—showed the effectiveness of metformin in diabetes prevention in terms of a reduction in the risk of progressing to type 2 diabetes mellitus in prediabetic individuals receiving the drug (pooled RR 0.58, indicating a 42% lower risk [136].

#### 4.1.3. Type 1 Diabetes

In the *Standards of Care* by the American Diabetes Association (ADA), metformin is also considered in paragraph “Non insulin treatment for type 1 diabetes” [137]. Meng et al. evaluated the effect of metformin in 1183 subjects with type 1 diabetes, demonstrating that metformin was associated with reductions in BMI (−1.14, 95% CI −2.05 to −0.24, *p* = 0.01), insulin requirements (−0.47, 95% CI −0.70 to −0.23, *p* = 0.0001), total cholesterol (−0.23, 95% CI −0.34 to −0.12, *p* < 0.0001), and low-density lipoprotein cholesterol (−0.20, 95% CI −0.29 to −0.11, *p* < 0.0001) in type 1 diabetic patients. No clear evidence indicated that metformin improved HbA1c, triglyceride, or high-density lipoprotein cholesterol levels [138]. The REMOVAL study focused on cardiovascular and metabolic effects of metformin in adults aged 40 years and older with type 1 diabetes of at least 5 years’ duration, demonstrating that progression of mean carotid intima-media thickness (IMT) was not significantly reduced with metformin, and that glycated hemoglobin was reduced on average over 3 years by metformin, but this was accounted for by a reduction at the 3-month timepoint that was not sustained thereafter. Bodyweight and LDL cholesterol were reduced with metformin over 3 years of treatment, and eGFR was increased. Finally, the insulin requirement was not reduced on average over 3 years [139].

### 4.2. Clinical Studies on Tumoral Conditions (“Anti-Proliferative Face”)

The promising in vitro and in vivo evidence regarding the antiproliferative activity of metformin has provided the rationale for the conduct of more than 130 clinical studies included up-to-date in the registry of clinicaltrials.gov. The aim of our paper is to focus on phase III clinical trials and major phase II trials when phase III is absent. The trials to evaluate metformin in oncology were prevalently performed in colorectal, breast, endometrial, and prostate cancer. Despite the increasing number of clinical studies, the data available concerning the potential antitumor effect of biguanide remain limited and inconsistent; therefore, more robust and large-scale trials are warranted to further investigate metformin administration in the neoplastic setting (Table 2).

#### Colorectal Cancer (CRC)

The antiproliferative efficacy of metformin explored in the therapeutic strategy for colorectal cancer (CRC) is still controversial. In a few prospective studies, the potential chemopreventive role of biguanide in CRC carcinogenesis was documented [142]. Hosono et al. evidenced that low-dose metformin treatment (250 mg daily) led to the inhibition of colonic epithelial proliferation and of colorectal aberrant crypt foci (ACF) in non-diabetic patients with ACF [160]. Furthermore in a phase III, multicenter, double-blind, and placebo-controlled RCT, metformin (250 mg daily)—administered for 1 year—was associated with a significantly lower prevalence of metachronous adenomas or polyps in non-diabetic patients at high risk of CRA relapse after polypectomy [161].

A similar chemopreventive effect was seen in a particular setting of patients with an increased risk of colonic polyps: acromegalic patients. In a recent exploratory cross-sectional study [162], we evaluated the prevalence of colonic polyps in acromegalic patients treated or not with metformin and explored its possible protective role against the development of colon polyps. We subdivided our cohort into patients with and without polyps. Only 24% of subjects with polyps were under metformin vs. 57% of patients without polyps, data confirmed by multivariate analysis (OR 0.224, 95% CI 0.065–0.770, *p* = 0.01). In our study, metformin was used as an antidiabetic treatment, so this finding could suggest that metformin therapy may counterbalance the dual risk factors of diabetes and acromegaly. These data suggest a potential protective role for metformin in a subset of individuals affected by this specific endocrine disease.

On the other hand, as usual in the scientific literature, other experiences yield different results. A phase IIa study involving non-diabetic and obese (BMI ≥ 30 kg/m^2^) subjects with a recent history of CRA undergoing metformin therapy 1000 mg twice daily for 12 weeks showed no reduction in rectal tissue pS6^Ser235/236^ or Ki67 immunostaining levels [163]. Similarly, in individuals already affected by CRC, the impact of the biguanide is still unclear and warrants further exploration. In a retrospective study conducted by Han et al. in 232 patients with rectal cancer who underwent curative resection after preoperative chemo-radiotherapy (CCRT), neoadjuvant metformin administration before CCRT resulted as a relevant factor in predicting tumor downstaging and good response rates of tumor regression grade [164]. Moreover, in a phase II trial, Miranda et al. described an overall modest activity of metformin 850 mg twice daily added to 5-fluorouralcil (5-FU) in 50 subjects with refractory CRC. In particular, 11 (22%) patients met the primary endpoint, which was disease control rate at 8 weeks, and among all patients, those with BMI ≥ 30 kg/m^2^ appeared to benefit more from the combined therapy. The median Progression Free Survival (mPFS) was 2 months and the median Overall Survival (mOS) was 7.9 months [140]. Less promising findings were detected in a phase II study by Akce et al., who documented that in patients with refractory microsatellite stable metastatic CRC undergoing combined treatment of metformin and nivolumab, no objective response was observed, hence the trial did not proceed with further enrollment; mOS and mPFS were respectively 5.1 and 2.3 months [141]. Accordingly, in a sub-study of the TOSCA trial involving individuals with high-risk stage II or stage III colon cancer undergoing 3 months versus 6 months of fluoropyrimidine–oxaliplatin adjuvant therapy, the addition of metformin impacted neither the OS nor the relapse-free survival (RFS), regardless of its dosage [142]. Similarly, no correlation between the biguanide and the survival outcomes, including OS, disease-free survival (DFS), and time to recurrence (TTR), was identified by Singh et al. in the setting of the phase III N0147 study involving 1958 patients with stage III colon cancer (CC) receiving adjuvant chemotherapy [165]. At present, there are two ongoing phase III RCTs enrolling individuals affected by CRC. In particular, Abdelhafeez et al. are investigating in stage IV CC the effect of metformin combined with the standard therapy FOLFOX/XELOX in comparison with their counterparts receiving only FOLFOX/XELOX. The primary outcomes concern the Disease Control Rate (DCR) and PFS [166]. Kim et al. are also conducting an open-label RCT to examine the impact of adjunctive metformin in patients with recurrent stage II high-risk and stage III colorectal cancer who have already undergone surgery and/or neo-adjuvant chemoradiation. The primary endpoint is the comparison of the 3-year DFS between metformin and non-metformin cohorts [167].

## 5. Breast Cancer (BC)

The potential antineoplastic role of metformin explored in the setting of both neo- and adjuvant therapy in patients affected by breast cancer is still unclear. In the phase III, double-blind and placebo-controlled RCT conducted in 3649 non-diabetic patients with BC, the addition of metformin to standard adjuvant therapy did not lead to a significant improvement in invasive disease-free survival [143]. Similarly, no significant survival benefits were detected in terms of PFS or response rate (RR) related to the combination of biguanide and chemotherapy in a study involving 107 non-diabetic patients with metastatic BC [144]. However, slightly higher OS and PFS, but with no statistical significance, were associated with metformin and adjuvant chemotherapy in the phase II trial of Salah. et al., enrolling 50 individuals with stage IV BC. The radiological response, instead, resulted significantly better in the metformin cohort in comparison to the placebo one [145]. Instead, more encouraging results were observed when metformin was administered in the neoadjuvant setting. In fact, in the METTEN phase II trial involving 79 individuals affected by HER2-positive BC bearing the rs11212617 C allele, metformin combined with neoadjuvant chemotherapy (anthracycline/taxane-based regimens) and ERBB2-targeted therapy (i.e., trastuzumab) was associated with a higher pathological complete response (pCR) compared to the non-metformin counterparts (81.2% vs. 35.3%, respectively) [146]. Moreover, Othman et al., in a phase II/III RCT, placebo-controlled study, involving 140 subjects with invasive non-metastatic BC, demonstrated that positive Her2 or negative estrogen receptor (ER) status seemed to predict pCR in the metformin-treated arm. In particular, ER-negative BC metformin therapy presented a significantly higher pCR compared to the control group (63.2% vs. 22.2%, respectively, *p* = 0.02). In HER-2-positive BC, a greater pCR was documented in the metformin arm, but no statistical significance was reached [147]. Similarly, Azazy et al. performed a phase II randomized and placebo-controlled trial in 60 non-diabetic patients with stage II–III BC and documented a higher pCR in the group receiving metformin (850 mg b.i.d.) and neoadjuvant chemotherapy without reaching any significance (*p* = 0.09) [148]. It is also documented that metformin has a positive impact when combined with neoadjuvant chemotherapy on the pCR rate, particularly in individuals with stage II-III triple-positive BC without diabetes and with a BMI ≥ 25 kg/m^2^ [149]. Discordantly, no improvement of pCR in patients affected by BC and metabolic abnormalities was observed in the phase II NeoMET study in the arm receiving both the biguanide and neoadjuvant chemotherapy (docetaxel, epirubicin, and cyclophosphamide) [150]. Also in the setting of metastatic or locally advanced, unresectable BC metformin treatment in conjunction with chemotherapy or hormone therapy tested in several phase II studies failed to present a relevant impact on the survival outcomes [151,152,153]. In conclusion, the anticancerogenic effect of biguanide in breast cancer remains controversial and warrants further investigation in robust RCTs. An ongoing phase III RCT is currently exploring the eventual role of metformin in preventing BC development in patients affected by atypical hyperplasia or in situ breast cancer. The results are going to be published in the forthcoming years and could also provide information regarding the potential chemopreventive activity of metformin [168].

## 6. Prostate Cancer

In prostate cancer, evidence shows inconsistent findings regarding the antitumor role of biguanide. In a meta-analysis comprising three phase III, double-blind, and place-controlled RCTs (AFFIRM, PREVAIL, and PROSPER) aimed at studying enzalutamide in castration-resistant prostate cancer (CRPC) patients, metformin treatment failed to present a significant impact on the survival outcomes [169]. Similarly, no benefits in terms of disease control rate (DCR) and OS were observed in the metformin-treated cohort in combination with enzalutamide compared to enzalutamide alone in SAKK 08/14, a phase II multicenter RCT involving 169 subjects with metastatic CRPC [154]. Also in TAXOMET, a phase II multicenter study comparing docetaxel plus metformin versus docetaxel plus placebo in metastatic CRPC, no meaningful benefits in terms of OS, PFS, ORR, and PSA response rate were documented [155]. Discordantly, a significantly higher prostate cancer-free survival (29 months vs. 20 months, *p* = 0.01), especially in individuals with high-risk localized disease or metastatic low tumor volume disease, was observed in MANSMED, a phase II RCT enrolling 124 patients with CRPC in the metformin plus androgen-deprivation therapy (ADT) arm versus ADT only [156]. Due to the insufficient data available, some randomized trials are currently ongoing. The MAST study, a phase III, double-blind, and placebo-controlled RCT, is evaluating the eventual impact of metformin in reducing disease progression in men affected by low-risk, localized prostate cancer on expectant management [170]. Another active study is STAMPEDE, a multi-arm and multi-stage phase II/III RCT already involving 11992 participants, aimed at exploring several therapeutic strategies, including metformin, in high-risk locally advanced and metastatic hormone-naïve prostate cancer [171]. In the forthcoming future, we are going to publish the findings of METAL, a phase IV placebo-controlled RCT finalized to investigate the role of biguanide in a neoadjuvant setting in early-stage prostate cancer patients [172].

## 7. Endometrial Cancer (EC)

There is limited data provided by prospective clinical trials regarding the potential antitumor effect of metformin in endometrial cancer. PREMIUM, a multi-center, placebo-controlled phase III RCT, conducted in 88 patients with atypical hyperplasia or endometrioid endometrial cancer, showed that neoadjuvant treatment with metformin (850 mg daily for 3 days and twice daily thereafter) for 1 to 5 weeks did not present a favorable impact on tumor proliferation. No differences were detected regarding the immunohistochemical expression of Ki-67, neither of the markers of PI3K-Akt-mTOR nor the insulin signaling pathway [157]. Conversely, Mitsuhashi et al., in a phase II, single-arm study enrolling 17 women with atypical endometrial hyperplasia (AEH) and 19 with EC, documented that metformin (750–2250 mg/day) was beneficial in preventing disease recurrence after medroxyprogesterone acetate (MPA) administration for fertility-sparing therapy, showing an 89% 3-year relapse-free survival (RFS) rate [158]. Such findings have provided the rationale for the commencement in 2019 of the FELICIA trial, a randomized phase IIb study, finalized particularly to investigate the adequate dose of metformin in addition to MPA in AEH and EC patients and identify the 3-year RFS rate. The trial results are still to be published [173]. In another ongoing phase III, randomized and placebo-controlled study, the role of biguanide is also being explored in the chemoprevention setting in obese (BMI ≥ 30 kg/m^2^) and hyperinsulinemic women affected by EC [174]. Other currently active trials are evaluating, in advanced or recurrent EC, the impact of metformin combined with everolimus and letrozole in a single-arm study [175], as well as metformin combined with paclitaxel plus carboplatin in a phase II/III placebo-controlled RCT [176].

## 8. Other Malignancies

Some interesting data on the antineoplastic efficacy of metformin derive from an Italian multicenter retrospective study on neuroendocrine tumors [177]. In this trial (PRIME-NET Study), the authors demonstrated a significantly longer PFS in diabetic patients with pancreatic neuroendocrine tumors (panNET) treated with metformin, everolimus, and/or somatostatin analogs, compared to diabetic patients with panNET treated with the same oncological therapy but with another antidiabetic agent instead of metformin. The PFS in the first group of diabetic patients with panNET was significantly longer, even compared to those of non-diabetic patients with panNET treated in the same manner, but without metformin. Clinical prospective data about the antineoplastic effect of metformin on other malignancies is still scarce and inconclusive. Regarding neuroendocrine neoplasms (NENs), METNET, a phase II single-arm trial, showed a modest antineoplastic activity of the biguanide (850 mg twice daily) in progressive metastatic well-differentiated NENs of gastroenteropancreatic (GEP) or pulmonary origin. In this study, 46% of the patients presented with DCR at 6 months and a mPFS of 6.3 months; however, this study enrolled only 28 subjects [159]. In consideration of the beneficial activity of metformin when added to everolimus and/or somatostatin analogs in NENs patients with diabetes, as evidenced by the preliminary findings of the PRIME-NET study [177], two phase II studies have been initiated. The MetNET-1 and MetNET-2 trials aim at evaluating the antiproliferative potential of the biguanide combined, respectively, with everolimus plus octreotide LAR in advanced well-differentiated pancreatic NENs or combined with lanreotide in well-differentiated GEP and lung NENs [178,179]. The results are still to be published and could provide the rationale for further exploration of metformin combined with target therapies in this cohort of patients. Preclinical evidence of the antiproliferative activity of metformin in other malignancies has also led to the design of some phase II or III RCTs, aimed at investigating the efficacy of the biguanide in lung cancer when added to tyrosine kinase inhibitors [180], in melanoma when combined with dacarbazine or with pembrolizumab [181,182], in hepatocellular carcinoma when administered with sorafenib [183], as well as in pancreatic cancer when combined with chemotherapeutic agents [184,185].

## 9. Conclusions and Future Research Directions

Metformin is an old drug with multiple target organs. Over the years, different mechanisms of action, such as the subcellular mechanisms involved in metabolism in cellular and tissue growth, not previously known, have been highlighted, arousing interest about the possibility of using this molecule in fields other than the one for which it is commonly used (T2DM). In addition to confirming the safety and effectiveness of metformin in reducing hba1c in the treatment of T2DM, this review confirms its role in situations where prediabetes are able to prevent the evolution of T2DM. In this last context, in our opinion, it should also be considered what it means to mask an evolution to T2DM and to establish the timing of complications checks and targets of other metabolic or clinical parameters (for example, blood pressure), which would otherwise be defined in the case of an evolution to T2DM. Data are not enough to allow important conclusions about the use of metformin in T1DM, where most likely the patient’s constitutional habit, beyond insulin deficiency, makes the difference in the sensitivity of organs to the response to metformin.

As regards the “antiproliferative face” of metformin, scientific knowledge to date suffers from a contrast between the encouraging in vitro results and the still inconclusive in vivo findings of some trials.

The most promising areas appear to be colon and breast cancers, although some very encouraging data derive from neuroendocrine neoplasms, though from retrospective data, on a large series of cases.

Therefore, further, more rigorous observational studies, or RCTs, should be designed properly, aimed at exploring the potential benefits of the biguanide in specific population subgroups accurately selected according to glycemic and metabolic status, cancer histological features, molecular profiling, and staging.

Considering the current data globally, it seems that the use of metformin in RCTs for a short period has not led to clinical benefits, while in some more prolonged experiences or in retrospective studies, in which patients had been taking the drug for other reasons for several years, the data are more suggestive of a certain benefit.

However, we have probably also considered this aspect, the duration of the patient’s exposure to the drug metformin, in order to evaluate the potential antiproliferative effect and its effectiveness.

Hence, proper data analysis methods should be implemented to avoid time-related biases that could have erroneously led to beneficial effects of metformin in cancer prevention and treatment in the non-randomized studies conducted since 2005, as highlighted by Hoi Yun Yu et al. [186]. In particular, several observational studies have been affected by the immortal time bias, which is defined as the time between cohort entry and metformin initiation, which could amplify the potential benefits of metformin by extending the exposure time to the biguanide and the survival time of the patient [186]. Therefore, a careful definition of drug exposure is another fundamental criterion for providing robust and accurate evidence.

The most interesting role of metformin could be its preventive impact against the development of neoplasia in populations at increased risk, rather than its implementation as an antiproliferative therapy in already diagnosed neoplasia.

As reported by Lord and Harris in a recent perspective article, metformin and its efficacy in cancer prevention have been underexplored in prospective studies [187].

Some data in the literature [161,162] provide a strong rationale for testing this drug in selected groups of patients, for example, acromegalic, obese patients, or insulin-resistant subjects [187].

Given the difficulty in obtaining results in the cancer prevention aspect, because the studies would require very large populations and very long follow-up, Lord and Harris emphasize that an attempt should instead be made to study the preventive power of metformin in those populations who have a higher risk of cancer, such as in some genetic syndromes or metabolic conditions [187].

An interesting translational study published a few years ago in the *Journal of Clinical Investigation* demonstrated an advantage in terms of increased cancer-free survival in a mouse model of Li–Fraumeni syndrome treated with metformin [188].

This effect was determined by the inhibition of mitochondrial respiration exerted by metformin and occurred in the same way both in the murine model and in vivo in patients affected by this syndrome. The authors therefore concluded that, having demonstrated that metformin was able to inhibit mitochondrial respiration in humans as well, it could be used to prevent the onset of cancer in this particular population of patients with Li–Fraumeni syndrome [188].

These data are also suggested by the literature available so far and will certainly be clarified in the future when the results of several currently ongoing studies are published [166,167,168,172,178,179].

For the future, instead, we should expect to find metformin increasingly under the magnifying glass of many authors as a possible association with immunotherapies. In fact, despite its high efficacy in many cancers, immunotherapy could be limited by the modulation of the tumor immune microenvironment.

A very recent review shows how metformin can be an important “booster” for immunotherapy, according to some preclinical studies (in vitro and animal models) and a few phase II studies on human tumor specimens derived from patients pretreated or not with metformin [189,190,191,192].

Some evidence from these studies shows how metformin, through mechanisms not yet fully elucidated, can modulate the tumor immune microenvironment, thus enhancing the response to immunotherapy.

In summary, we do not yet have a conclusive vision of the potential applications of metformin in the oncology field. At this moment, the evidence about its effectiveness in the treatment of cancers is disappointing; however, the data regarding its role in preventing cancer are intriguing and worthy of further exploration with ongoing trials and others that may clarify the “anti-neoplastic face” of metformin in the future.

## Figures and Tables

**Figure 1 cancers-16-01287-f001:**
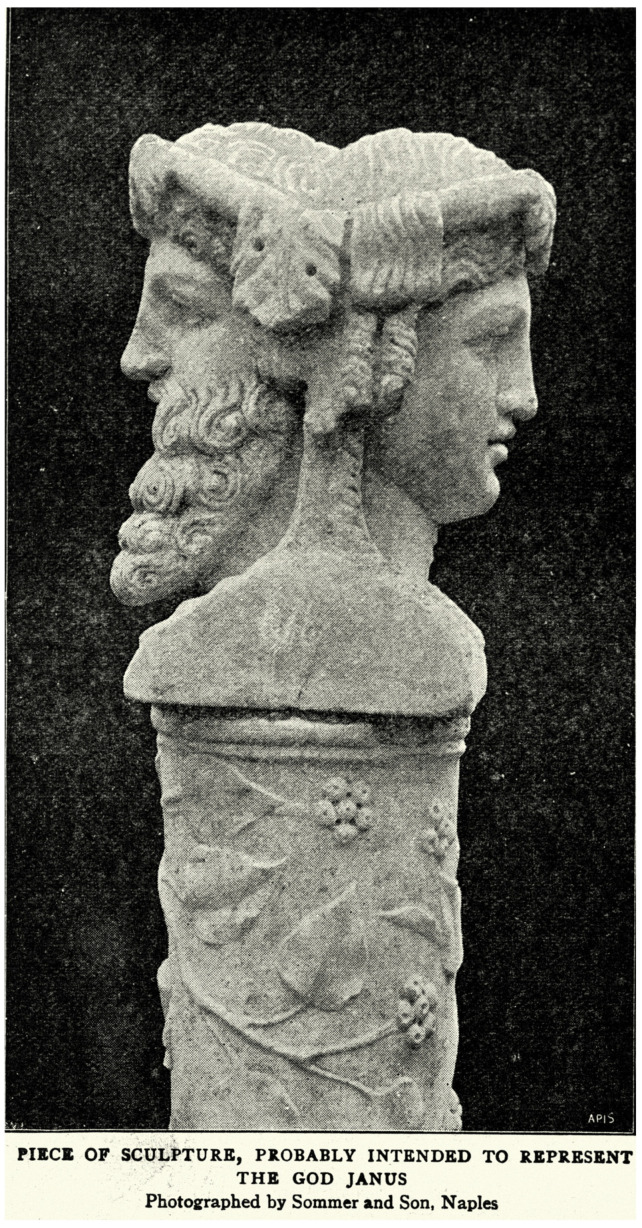
*Ianus Bifrons* sculpture (https://iStock.com/ accessed on 22 May 2023).

**Figure 2 cancers-16-01287-f002:**
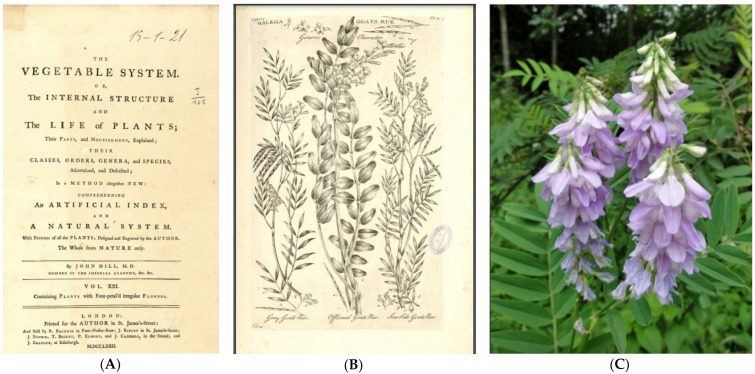
(**A**,**B**) Frontspiece of Volume XXI of John Hill’s huge work and graphic reproduction of *Galena officinalis*, p54, 1772 [2]. (**C**) Photo of *Galena officinalis* by Peter Smith, Aylestone Meadows, 10 July 2013, www.natarespot.org.uk (accessed on 22 May 2023).

**Figure 3 cancers-16-01287-f003:**
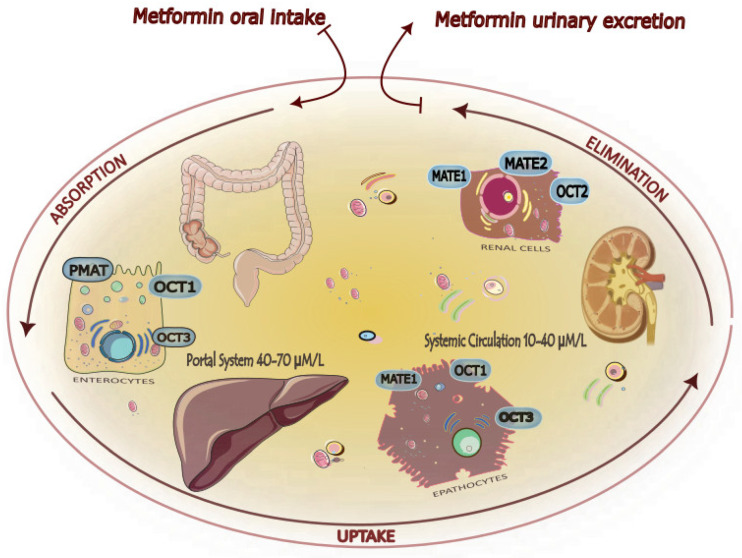
*Absorption*, *uptake*, and *elimination* of metformin. After oral intake, absorption in the gastrointestinal tract is mediated by specific molecular transporters that allow a drug concentration in the portal system of 40–70 µmol/L, higher than that in systemic circulation (10–40 µmol/L). Metformin is excreted unchanged in the urine, and active tubular secretion in the kidney is the main route of drug elimination. PMAT: plasma membrane monoamine transporter (expressed within the apical membranes of enterocytes in the small intestine, and variants are associated with poor tolerance in subjects affected by diabetes mellitus). OCT1/2/3: organic cation transporter. MATE1/2: multidrug and toxin extrusion proteins.

**Figure 4 cancers-16-01287-f004:**
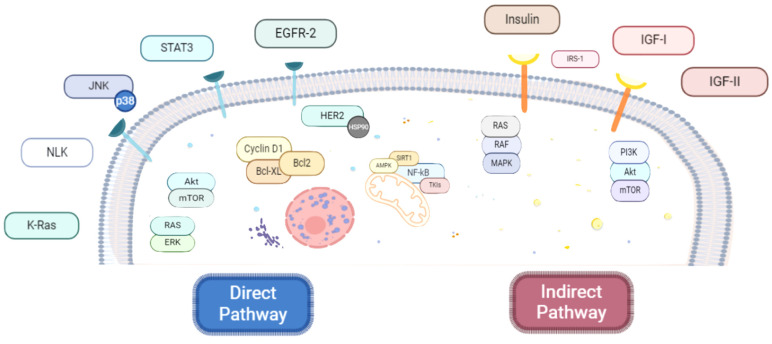
Direct (**left**) and indirect (**right**) pathways involved in the molecular mechanisms of the antineoplastic role of metformin. For specific actions of subcellular signals, see the text.

**Table 1 cancers-16-01287-t001:** Comparisons between guanine, guanidine, and biguanides (metformin, phenformin, and buformin). LogP: *octanol-water partition coefficient* (Log P is positive for lipophilic and negative for hydrophilic substances or species).

Name	Chemical Formula	Chemical Structure	Features and Origin	Solubility LogP	Binding to Mithocondrial Membranes	Tissue of Anaerobic Glycolysis	Metabolism	Risk of Lactic Acidosis (Events for 1000 Subjects/Year)
**GUANINE**	C_5_H_5_N_5_O	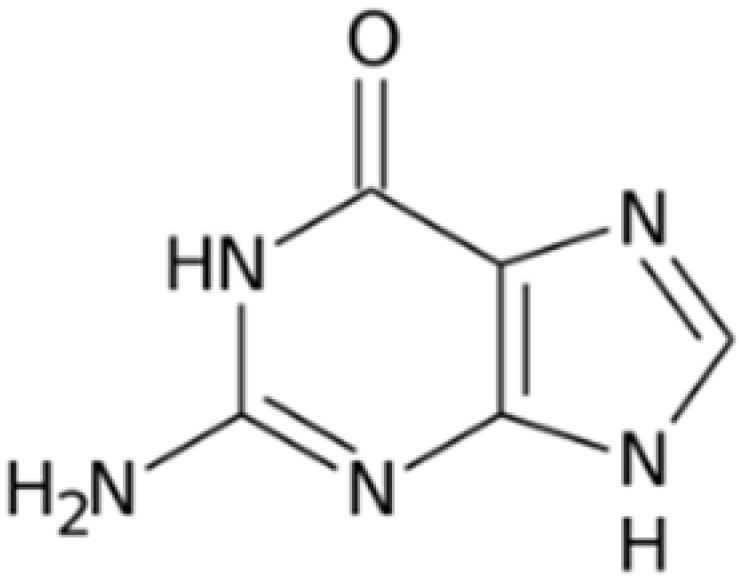	In 1844, the German chemist Julius Bodo Unger obtained it as a mineral formed from the excreta of sea birds (*guano*).	Insoluble in water				
**GUANIDINE**	HNC(NH_2_)_2_	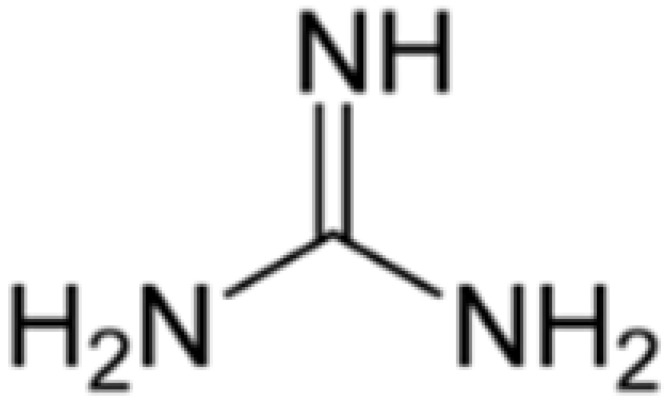	It is a strong base, obtained from natural source, via the oxidative degradation of guanine	Soluble in water and ethanol −1.7				
**METFORMIN**	C_4_H_11_N_5_	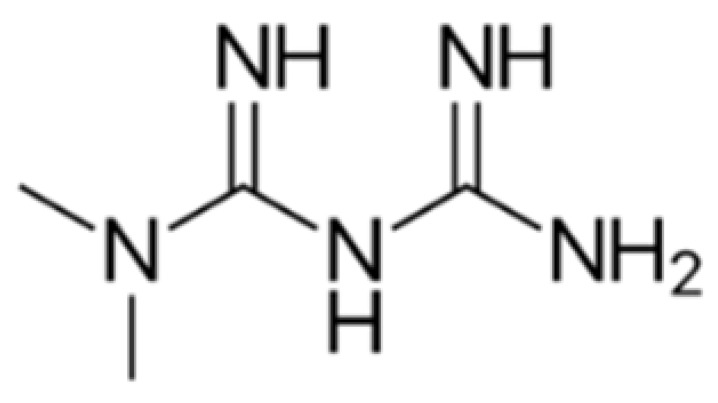	N,N-dimethylamine guanylguanidine (chemical syntesis)	More hydrophilic −1.43	*Weaker*	Mostly intestinal tissue exposed to high drug concentration	Not metabolized, eliminated unchanged	0.03–0.09
**PHENFORMIN**	C_10_H_15_N_5_	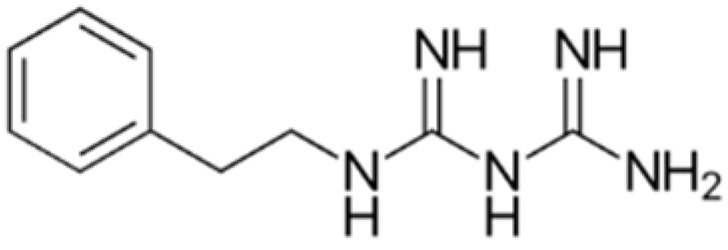	It was developed in 1957 by Ungar, Freedman, and Shapiro	More lipophilic −0.83	*Stronger*	More generalized, including muscle	Not metabolized, eliminated unchanged	0.40–0.90
**BUFORMIN**	C_6_H_15_N_5_	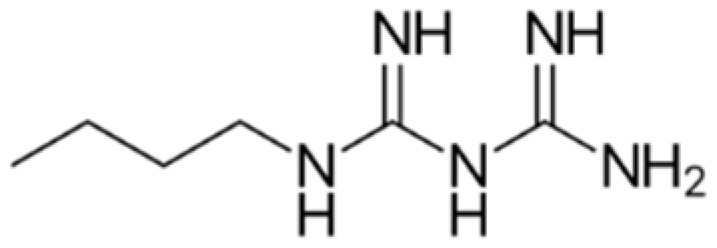	It is a strong base, and is freely soluble in water, methanol, and ethanol	Intermediate −1.20	*Stronger*	More generalised, including muscle	Not metabolized, eliminated unchanged	>0.1

**Table 2 cancers-16-01287-t002:** Type of study, intervention/treatment, patients, and outcomes in main clinical trials in different neoplasms (colorectal cancer, breast cancer, prostate cancer, and GEP-NET).

Trial Title	Tumor	Type of Study	Intervention/Treatment	Patients, n	Primary Outcome (s)	Results	References
Phase 2 Trial of Metformin Combined With 5-Fluorouracil in Patients With Refractory Metastatic Colorectal Cancer	Colorectal cancer	Single-arm, phase II study	Metformin 850 mg bid plus 5-FU 425 mg/m^2^ and leucovorin 50 mg i.v. weekly	50	DCR at 8 weeks	Modest activity of metformin plus 5-FU with major benefits observed in patients with BMI ≥ 30 kg/m^2^ 11 (22%) patients presented DCR at 8 weeks with mPFS = 5.6 months and mOS = 16.2 months	Miranda et al., 2016 [140]
Phase II trial of nivolumab and metformin in patients with treatment-refractory microsatellite stable metastatic colorectal cancer.	Colorectal cancer	Single-arm, phase II study	Metformin 1000 mg bid plus Nivolumab 480 mg i.v. every 4 weeks	24	ORR	No ORR was observed; the study did not proceed with further enrollment	Akce et al., 2023 [141]
Impact of Metformin Use and Diabetic Status During Adjuvant Fluoropyrimidine-Oxaliplatin Chemotherapy on the Outcome of Patients with Resected Colon Cancer: A TOSCA Study Subanalysis	Colon cancer	Substudy of phase III TOSCA trial	Metformin plus adjuvant fluoropyrimidine-oxaliplatin	3759	OS and RFS	No impact on OS or RFS	Vernieri et al., 2019 [142]
Effect of Metformin vs. Placebo on Invasive Disease-Free Survival in Patients With Breast Cancer: The MA.32 Randomized Clinical Trial	Breast cancer	Phase III, placebo-controlled, double-blind RCT	Drug: metformin 850 mg/day for 4 weeks, then 850 mg bid for 5 years plus adjuvant standard breast cancer treatment Control arm: placebo plus adjuvant standard breast cancer treatment	3649	Invasive disease-free survival	No significant improvement of the invasive disease-free survival	Goodwin et al., 2022 [143]
Efficacy of Metformin as Adjuvant Therapy in Metastatic Breast Cancer Treatment	Breast cancer	Prospective, placebo-controlled RCT	Drug: metformin 500 mg bid plus adjuvant CT Control arm: adjuvant CT	107	PFS and RR	No significant benefits on PFS and RR	Essa et al., 2022 [144]
Metformin as an Adjuvant Treatment in Non-Diabetic Metastatic Breast Cancer	Breast cancer	Phase II RCT	Drug: metformin 1000 mg bid plus adjuvant CT Control arm: CT	50	OS and PFS	Metformin group vs. control group presented no significant improvement of OS and PFS, but higher radiological response (*p* = 0.002)	Salah et al., 2021 [145]
The C Allele of ATM rs11212617 Associates With Higher Pathological Complete Remission Rate in Breast Cancer Patients Treated With Neoadjuvant Metformin (METTEN)	Breast cancer	Phase II, open-label, multicenter RCT	Drug: metformin 850 mg bid for 24 weeks plus anthracycline/taxane-based CT and trastuzumab Control arm: neoadjuvant anthracycline/taxane-based CT and trastuzumab	79	PCR	Metformin group was associated with a higher PCR than control group	Cuyàs et al., 2019 [146]
Neoadjuvant chemotherapy with or without metformin in invasive nonmetastatic breast cancer. Randomized controlled trial	Breast cancer	Phase II/III RCT	Drug: metformin 850 mg/day, then 850 mg bid plus neoadjuvant CT Control arm: neoadjuvant CT	140	Tumor RR	Metformin plus neoadjuvant CT was associated with higher PCR, especially in ER-negative BC (63.2% vs. 22.2%, *p* = 0.02) and ER-negative BC (50% vs. 34.6%, *p* = 0.3)	Othman et al., 2023 [147]
Metformin with neoadjuvant chemotherapy in stage II-III breast cancer: A phase II clinical trial.	Breast cancer	Phase II RCT	Drug: metformin 850 mg bid plus neoadjuvant CT Control arm: neoadjuvant CT	60	OPR	A higher pCR was observed in the metformin group, but without statistical significance (*p* = 0.09)	Azazy et al., 2020 [148]
The effect of metformin when combined with neoadjuvant chemotherapy in breast cancer patients.	Breast cancer	Prospective study	Drug: metformin 850 mg bid plus neoadjuvant CT Control arm: neoadjuvant CT	59	PCR	The addition of metformin may improve PCR particularly in individuals with triple-positive BC and BMI ≥ 25 kg/m^2^	El-Khayat et al., 2021 [149]
Neoadjuvant docetaxel, epirubicin, and cyclophosphamide with or without metformin in breast cancer patients with metabolic abnormality: results from the randomized Phase II NeoMET trial	Breast cancer	Phase II RCT	Drug: metformin 850 mg/day for the first cycle, then 850 mg bid plus neoadjuvant CT (TEC) Control arm: neoadjuvant CT (TEC)	92	PCR	No improvement of PCR	Huang et al., 2023 [150]
A phase II randomized clinical trial of the effect of metformin versus placebo on progression-free survival in women with metastatic breast cancer receiving standard chemotherapy	Breast cancer	Phase II, double-blind, RCT	Drug: metformin 850 mg bid plus CT Control arm: placebo plus CT	40	PFS	No significant impact on PFS	Pimentel et al., 2019 [151]
Metformin plus chemotherapy versus chemotherapy alone in the first-line treatment of HER2-negative metastatic breast cancer. The MYME randomized, phase 2 clinical trial.	Breast cancer	Phase II RCT	Drug: metformin 2000 mg/day plus CT Control arm: CT	122	PFS	No significant impact on PFS	Nanni et al., 2019 [152]
A randomized phase II study of aromatase inhibitors plus metformin in pre-treated postmenopausal patients with hormone receptor positive metastatic breast cancer.	Breast cancer	Phase II RCT	Drug: metformin 500 mg bid plus aromatase inhibitor (exemestane 25 mg/d or letrozole 2.5 mg/d) Control arm: aromatase inhibitor (exemestane 25 mg/d or letrozole 2.5 mg/d)	60	PFS	No significant impact on PFS	Zhao et al., 2017 [153]
SAKK 08/14—IMPROVE Investigation of metformin in patients with metastatic castration-resistant prostate cancer (mCRPC) in combination with enzalutamide vs. enzalutamide alone. A randomized, open label, phase II trial.	Prostate cancer	Phase II, open-label, multicenter RCT	Drug: metformin 850 mg bid plus enzalutamide 160 mg Control arm: enzalutamide 160 mg	169	DCR at 15 months	No benefits on DCR	Rothermundt et al., 2022 [154]
TAXOMET: A French prospective multicentric randomized controlled phase II study comparing docetaxel plus metformin versus docetaxel plus placebo in mCRPC.	Prostate cancer	Phase II, placebo-controlled, multicenter RCT	Drug: metformin 850 mg bid plus docetaxel 75 mg/m^2^ every 21 days plus prednisone 5 mg bid Control arm: placebo plus docetaxel 75 mg/m^2^ every 21 days plus prednisone 5 mg bid	99	PSA response ≥ 50% from baseline	No significant benefits on PSA response, PFS, OS or ORR	Martin et al., 2021 [155]
Repurposing metformin as anticancer drug: Randomized controlled trial in advanced prostate cancer (MANSMED)	Prostate cancer	Phase II RCT	Drug: metformin plus standard of care Control arm: standard of care	124	CRPC-FS	A significantly higher CRPC-FS (29 months vs. 20 months, *p* = 0.01) was observed in the metformin group, especially in individuals with high-risk localized disease or metastatic low tumor volume disease	Alghandour et al., 2021 [156]
PRE-surgical Metformin In Uterine Malignancy (PREMIUM): a Multi-Center, Randomized Double-Blind, Placebo-Controlled Phase III Trial	Endometrial cancer or AEH	Phase III, multicenter, double-blind RCT	Drug: neoadjuvant metformin 850 mg/d for 3 days, then 850 mg bid for 1 to 5 weeks Control arm: placebo for 1 to 5 weeks until surgery	88	Post-treatment IHC expression of Ki-67	No differences in Ki67 expression were detected	Kitson et al., 2019 [157]
Phase II study of medroxyprogesterone acetate plus metformin as a fertility-sparing treatment for atypical endometrial hyperplasia and endometrial cancer.	Endometrial cancer or AEH	Single-arm, phase II trial	Drug: metformin (750–2250 mg/day) plus medroxyprogesterone acetate 400 mg/day	17 with AEH and 19 with EC	RFS after remission	Beneficial effect of metformin in inhibiting disease relapse (3-year RFS rate = 89%)	Mitsuhashi et al., 2016 [158]
METNET: a phase II trial of metformin in patients with well-differentiated neuroendocrine tumours.	GEP-NET or pulmonary NET	Single-arm, phase II trial	Drug: metformin 850 mg bid	28	DCR at 6 months	Modest antineoplastic activity of metformin in well-differentiated GEP or lung NET 26 patients had progression, 13 (46%) of whom presented DCR at 6 months and mPFS 6.3 months	Glasberg et al., 2022 [159]

Abbreviations: RCT: randomized clinical trial, bid: twice a day, 5-FU: 5-flurouracil, i.v.: intravenous, DCR: disease control rate, BMI: body mass index, PFS: progression-free survival, mPFS: median PRS, OS: overall survival, mOS: median OS, ORR: overall response rate, RFS: relapse-free survival, CT: chemotherapy, RR: response rate, PCR: pathological complete response, OPR: overall pathological response, BC: breast cancer, TEC: TEC docetaxel 75 mg/m^2^, epirubicin 75 mg/m^2^, and cyclophosphamide 500 mg/m^2^, d1, q3w; PSA: prostate-specific antigen, CRPC-FS: castration-resistant prostate cancer-free survival, AEH: atypical endometrial hyperplasia, IHC: immunohistochemical, GEP: gastreoenteropancreatic, and NET: neuroendocrine tumor.

## Data Availability

The data presented in this study are available in this article.

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
