# Peer review of "Ianus Bifrons: The Two Faces of Metformin"

_cancers, 2024, doi:10.3390/cancers16071287_

Round 1
Reviewer 1 Report
Comments and Suggestions for Authors
In this manuscript, the authors summarized current knowledge regarding metformin and cancer. They started with the historical background, accordingly, described the established and potential mechanisms of metformin. They also mentioned about ongoing clinical study to assess the effects of metformin on cancer as an adjuvant therapy. Taken together, the manuscript is well-known and comprehensive.
Comments:
1) "Complex I inhibition-dependent mechanism" and "mGPDH-dependent and Complex IV inhibition-dependent mechanism" should be described for more detail but not bullet points.
2) Figure that shows signaling pathways regulated by metformin may be added. This would be useful for readers to understand anti-cancer properties of metformin.
Author Response
We are very grateful for your deep appreciation of our work, and we thanks for suggestions to further improvement of the manuscript.
In order to your advice we have:
- Insert more details to explain the molecular mechanisms involved in glucose homeostasis about the “Complex I inhibition-dependent mechianism and mGPDH-dependent and Complex IV inhibition-dependent mechanism”, in paragraph 3: “Molecular mechanisms involved in metformin action”
- Add a specific figure (Fig.4) displaying signaling pathways to better understand anticancer properties of metformin
Reviewer 2 Report
Comments and Suggestions for Authors
The primary issue with the manuscript is a lack of novelty. The authors reach the conclusion that metformin is efficacious in Type 2 diabetes but that there is no conclusive evidence for its use as a cancer treatment. Neither of these conclusions adds to the field significantly. There are already several reviews that address the inconclusive nature of the use of metformin as a cancer therapeutic that more cogently summarize and evaluate the literature. For example, Lord and Harris (2023) directly question the value of the continued study of metformin as a potential cancer therapeutic, given the amount of study that occurred without translation.
Lord, S. R., & Harris, A. L. (2023). Is it still worth pursuing the repurposing of metformin as a cancer therapeutic? British Journal of Cancer, 128(6), 958–966. https://doi.org/10.1038/s41416-023-02204-2
The purpose of this review is not clear. The authors attempt to address metformin’s possible multifaceted therapeutic potential with the Ianus metaphor but conclude that it is only clearly effective in its approved clinical application.
The writing is disjointed and difficult to follow. The authors state the purpose directly in the simple summary, and from that statement, it’s clear that too many disparate directions were pursued.
Comments on the Quality of English Language
Extensive English language editing is required.
Author Response
Since 2005, when the first observational study documenting beneficial effects of metformin in reducing cancer incidence, there has been an increasing interest in investigating the potential antineoplastic effects of metformin in oncological patients.Over the last years, several studies have been reported in literature.
By the kind invitation of the Guest Editors, we conducted not a systematic but a narrative review on the requested topic, regarding the antimetabolic and antiproliferative aspects associated with metformin in Special Issue “Cancer and Diabetes: what connections between them”.
We would like to thank the Reviewer_2 for critical judgment and for the paper suggested, which was highly appreciated also due to the concordance with some of our conclusions, especially regarding a major role of metformin in the preventive setting rather than the impact on the survival outcomes in cancer patients, as well as the difficult evaluation of the drug efficacy in the studies conducted.
Therefore, our aim was to present only the most important fase 2 or fase 3 randomized trials to provide more accurate evidence in different cancer types.
Thereby, we further have commented such papers extensively, and all comments were inserted in our review. The related modifications are reported from line 627 to line 645.
Moreover, we have also added and commented another reference related to the possible biases associated with the observational studies regarding metformin in oncological patiens. Such modifications are reported from line 597 to line 600 and from line 609 to line 617.
We have modified conclusive lines accordingly, from line 654 to line 658.
Finally, in order to make our review more informative, we have added to the table another column with the results of the trials, and have insert in "References" Section other specific published scientific papers.
Thank you so much for your suggestions to improve our work.
Reviewer 3 Report
Comments and Suggestions for Authors
Dear authors,
You present here a review regarding the potential of an old drug, namely metformin, a biguanide well-known and used in the treatment regimen of type 2 diabetes.
I liked the originality of your approach, comparing metformin with a roman god, with two faces, therefore presenting the classic use of this drug as an antidiabetic and the new "face" as an antiproliferative agent.
The references are well-chosen and in agreement with the subject.
There are some English language aspects that I want to detail:
- add "the" in "In the Nineteenth..." (line 87), in "after the oral intake.." (line 2, page 14), in "of the oral absorption", in "due to the low intestinal..." (line 6, page 14)
- "the" plasma protein (line 23, page 14)
- add "d" in "adverse effect" (line 32, page 14)
- reduce the size of Figure 3
- eliminate the phrase in line 36-38 in page 15, they are already written in text
- add "the" in lines 81 (after the intravenous)
- correct in line 83: "to the hypoglycemic effect of the drug"
- you say Figure 4 in line 108 in page 17. There is no Figure 4 in the paper. Maybe you are referring to Figure 3?
- there are some empty spaces between words, such in lines 110, 112, 113, 119, 124, 314,etc
- correct in line 121: "induces", "of the mitochondrial"
- correct in line 132: "acts", "via the Complex", "with the following"
- in line 138, page 17, you want to say "is raised through..."?
- correct "hemoglobin" in line 344, page 22
- correct "infection" in line 348, page 22
- correct "gastrointestinal" in line 349, page 22
Comments on the Quality of English LanguageThere are some English language aspects that I want to detail:
- add "the" in "In the Nineteenth..." (line 87), in "after the oral intake.." (line 2, page 14), in "of the oral absorption", in "due to the low intestinal..." (line 6, page 14)
- "the" plasma protein (line 23, page 14)
- add "d" in "adverse effect" (line 32, page 14)
- reduce the size of Figure 3
- eliminate the phrase in line 36-38 in page 15, they are already written in text
- add "the" in lines 81 (after the intravenous)
- correct in line 83: "to the hypoglycemic effect of the drug"
- you say Figure 4 in line 108 in page 17. There is no Figure 4 in the paper. Maybe you are referring to Figure 3?
- there are some empty spaces between words, such in lines 110, 112, 113, 119, 124, 314,etc
- correct in line 121: "induces", "of the mitochondrial"
- correct in line 132: "acts", "via the Complex", "with the following"
- in line 138, page 17, you want to say "is raised through..."?
- correct "hemoglobin" in line 344, page 22
- correct "infection" in line 348, page 22
- correct "gastrointestinal" in line 349, page 22
Author Response
We are very gratefulfor your genuine appreciation of our proposed paper, mainly about the mythological comparison between the ancient roman god Ianus and the old molecule metformin.
We further thanks for your accurate and patient contribute to ameliorate the flow of text, indentifying several points of the manuscripts that we have corrected.